# Long-Term Pioglitazone Treatment Has No Significant Impact on Microglial Activation and Tau Pathology in P301S Mice

**DOI:** 10.3390/ijms241210106

**Published:** 2023-06-14

**Authors:** Lea Helena Kunze, François Ruch, Gloria Biechele, Florian Eckenweber, Karin Wind-Mark, Lina Dinkel, Paul Feyen, Peter Bartenstein, Sibylle Ziegler, Lars Paeger, Sabina Tahirovic, Jochen Herms, Matthias Brendel

**Affiliations:** 1Department of Nuclear Medicine, University Hospital of LMU Munich, Marchioninistraße 15, 81377 Munich, Germany; 2German Center for Neurodegenerative Diseases (DZNE) Munich, Feodor-Lynen-Str. 17, 81377 Munich, Germany; 3Department of Radiology, University Hospital of LMU Munich, Marchioninistraße 15, 81377 Munich, Germany; 4Munich Cluster for Systems Neurology (SyNergy), Ludwig Maximilian University of Munich, 81377 Munich, Germany; 5Center for Neuropathology and Prion Research, LMU Munich, Feodor-Lynen-Str. 23, 81377 Munich, Germany

**Keywords:** microglia, pioglitazone, TSPO-PET

## Abstract

Neuroinflammation is one disease hallmark on the road to neurodegeneration in primary tauopathies. Thus, immunomodulation might be a suitable treatment strategy to delay or even prevent the occurrence of symptoms and thus relieve the burden for patients and caregivers. In recent years, the peroxisome proliferator-activated receptor γ (PPARγ) has received increasing attention as it is immediately involved in the regulation of the immune system and can be targeted by the anti-diabetic drug pioglitazone. Previous studies have shown significant immunomodulation in amyloid-β (Aβ) mouse models by pioglitazone. In this study, we performed long-term treatment over six months in P301S mice as a tauopathy model with either pioglitazone or placebo. We performed serial 18 kDa translocator protein positron-emission-tomography (TSPO-PET) imaging and terminal immunohistochemistry to assess microglial activation during treatment. Tau pathology was quantified via immunohistochemistry at the end of the study. Long-term pioglitazone treatment had no significant effect on TSPO-PET, immunohistochemistry read-outs of microglial activation, or tau pathology levels in P301S mice. Thus, we conclude that pioglitazone modifies the time course of Aβ-dependent microglial activation, but does not significantly modulate microglial activation in response to tau pathology.

## 1. Introduction

Neuroinflammation is a common feature of many neurological diseases [1,2]. In the brain, inflammation is caused by the inherent immune system of the brain, which consists of astrocytes and microglia, both glial cells that support the neuronal network [3]. Microglia form a distinct population among myeloid immune cells of the organism [4] and they are critical to maintaining the physiological state of the brain by releasing neurotrophic factors and removing possibly neurotoxic debris such as pathological protein aggregates [2,3,5,6]. However, dysregulation leading to chronic inflammation can contribute to neurodegeneration [1,3,7]. Thus, along the disease progression, proinflammatory cytokines might promote amyloid-β (Aβ) accumulation in APPPS1 mice [6]. In a tauopathy mouse model, neurodegeneration was found to be driven by activated microglia rather than by the spread of aggregated tau itself [8]. Hence, the modulation of the immune system to temper detrimental phenotypes is an important consideration in the development of therapeutic strategies concerning neurodegenerative diseases.

One well-established option to achieve immunomodulation is by targeting the peroxisome proliferator-activated receptor γ (PPARγ) through pioglitazone [9,10]. PPARγ is a transcription factor [11] involved in many different physiological functions, and amongst others are cell differentiation, cell death, glucose metabolism, and insulin sensitivity [11,12,13]. Moreover, it has been shown that PPARγ modulates multiple genes associated with the immune response, some also being dysregulated in late-onset Alzheimer’s disease [14,15]. Pioglitazone is a well-known PPARγ agonist and an approved drug for the treatment of type 2 diabetes [11,16,17] that enhances PPARγ expression [17,18]. In earlier studies, mostly in Aβ mouse models, pioglitazone has been shown to be a promising treatment for Alzheimer’s disease, ameliorating both the pathology as well as the cognition in animal models [9,17,18,19,20,21,22,23,24,25].

In this study, we investigated the efficacy of long-term pioglitazone treatment to modulate chronic inflammation in a mouse model of tauopathy. Based on previous findings in Aβ mouse models, we tested the hypothesis that decreased microglial activation is detectable with serial 18 kDa translocator protein positron-emission-tomography (TSPO-PET) in pioglitazone-treated P301S mice, a well-established mouse model for tauopathy, to add to previous studies in Aβ mouse models [26,27,28,29]. TSPO expression is increased in activated microglia, and PET-tracers targeting this protein have been shown to be suitable tools to monitor neuroinflammation [26,30,31].

Furthermore, we aimed to validate the TSPO-PET results using ionized calcium-binding adapter molecule 1 (Iba1) and Cluster of Differentiation 68 (CD68) immunohistochemistry and we performed AT8 immunohistochemistry to test for alterations of phosphorylated tau accumulation after long-term pioglitazone treatment.

## 2. Results

### 2.1. Pioglitazone Treatment Has No Significant Effect on Serial TSPO-PET Signals in P301S Mice

The TSPO-PET results of nucleus accumbens normalization are visualized in Figure 1. Mixed-effect models of the different groups revealed significant differences in TSPO-signals between P301S and wild-type mice in all target areas, including the brainstem and cerebellum (*p* < 0.0001), hippocampus (*p* = 0.0014), and cortex (*p* = 0.0273). However, no significant differences could be observed for the treatment status (pioglitazone vs. placebo) within P301S and wild-type cohorts, neither as a general treatment effect nor with Tukey’s multiple comparisons test, except for in wild-type mice at eight months of age (*p* = 0.0451). Similar results were found for SUV normalized data, myocardium adjusted SUV, or %ID (Appendix A).

In particular, the brainstem TSPO-PET signal showed a time-dependent increase in P301S mice treated with pioglitazone relative to baseline (7.3 months: +21%, *p* = 0.01; 8.2 months: +23%, *p* = 0.002). However, placebo-treated P301S mice indicated a similar TSPO-PET signal increase in the brainstem relative to baseline (7.3 months: +22%, *p* = 0.006; 8.2 months: +23%, *p* = 0.003). Similar TSPO-PET signal increases in P301S mice with and without pioglitazone treatment were observed for the cerebellum. Again, the comparison of TSPO-PET results with different normalization approaches did not indicate long-term pioglitazone-related treatment effects on the rate of TSPO-PET change over time in P301S or wild-type mice (Appendix A).

### 2.2. Pioglitazone Treatment Has No Significant Effect on the Abundance of Tau-Positive Cells in P301S Mice

Immunohistochemical stainings with AT8 allowed for the quantification of tau-positive cells in the cortex and the brainstem of P301S mice treated with pioglitazone or placebo. On average, P301S mice receiving placebo had 143.0 ± 42.3 tau-positive cells in 0.014 mm^3^ of the cortex and 99.0 ± 33.2 tau-positive cells in 0.011 mm^3^ of the brainstem, whereas P301S mice treated with pioglitazone had 120.9 ± 45.9 (−15%) tau-positive cells in the cortex and 112.8 ± 39.1 (+14%) tau-positive cells in the brainstem (Figure 2). None of these comparisons reached statistical significance (cortex: *p* = 0.35, brainstem: *p* = 0.44).

### 2.3. Pioglitazone Has No Significant Effect on Iba1 and CD68 Expression in P301S Mice

The immunohistochemical assessment of Iba1 and CD68 expression in 0.002 mm^3^ of the cortex and the brainstem of P301S mice did not show significant differences between pioglitazone treated mice compared to placebo (Figure 3). While a trend towards lower Iba1 and CD68 reactivity was observed for the cortex of pioglitazone treated P301S mice compared to placebo (*p* = 0.26 and *p* = 0.08, respectively), the brainstem indicated a similar abundance of both markers between treatment and placebo groups (*p* = 0.51 and *p* = 0.50, respectively).

## 3. Discussion

Pioglitazone is considered to be a potential treatment for neurodegenerative diseases, as several preclinical trials have shown beneficial effects on different disease-related hallmarks. In particular, pioglitazone enhanced memory and learning, as well as ameliorated behavior [19,20,32], although in the study of Seok et al. (2019) [21] only at lower doses. On a mechanistic level, the activation of PPARγ by pioglitazone hindered the emergence and promoted the clearing of pathological protein aggregation, such as of Aβ [9,17,18,22,23] and tau [23], while some effects were dose- and area-dependent [21]. Moreover, the drug contributed to immunosuppression through acting on PPARγ [17,18,22]. Hence, it reduced the number of reactive astrocytes and microglia in APP/PS1 mice, thereby showing anti-inflammatory action while still promoting phagocytosis of Aβ-plaques [9]. Additionally, more recent studies have shown a positive effect of pioglitazone treatment on behavior [24] as well as a reduction in inflammation as measured by the attenuation of the TSPO-PET signal in PS2APP and APP^NL-G-F^ mice [25].

Of note, there have also been some preclinical studies showing no effect or even detrimental effects of pioglitazone on the brain. Thus, despite improved cerebral blood flow, Aβ pathology has kept progressing in different amyloid mouse models [33,34], and these studies have not found any change in memory and cognition [33,34]. Late treatment initiation in these studies needs to be considered, with treatment starting at 10 months of age [33] and 10–12 months of age [34].

Given the body of preclinical evidence, several clinical trials have investigated the effectiveness of pioglitazone in humans. Generally, diabetes patients were indicated to have a lower risk and later onset of dementia when treated with pioglitazone [35,36]. However, pioglitazone treatment was unable to improve cognition or alter the age of onset of mild cognitive impairment in non-diabetic volunteers [37]. Moreover, one clinical trial showed no adverse, but also no beneficial treatment effects of pioglitazone in patients with probable Alzheimer’s Disease [38], and another recent phase III clinical trial with individuals with high risk for Alzheimer’s Disease was terminated early due to the inefficacy of pioglitazone treatment [39].

Since most preclinical studies focused on Aβ mouse models, we intended to investigate pioglitazone in the presence of tau-pathology-related neuroinflammation and used the previously characterized P301S mouse model [26]. As expected, we were able to reproduce a significant time-dependent increase in TSPO-PET signals in P301S mice. However, we could not observe a significant impact of pioglitazone treatment on the rate of change in serial TSPO-PET results, nor in the immunohistological assessment of tau-positive neurons and the microglia markers Iba1 and CD68. Thus, our results in a tau mouse model are in contrast to previous studies that found a decrease in microglial activation after pioglitazone treatment in APPV717I [22], A/T [34], PS2APP, and APPNL-G-F Aβ mouse models [25]. We conclude that the difference in the underlying neuropathology might be the reason for the effectiveness of the PPARγ-related modulation of microglial activation. We speculate that this could be one important reason for the failure of pioglitazone in clinical trials of Alzheimer’s disease, because Alzheimer’s disease comprises an Aβ-plaque-mediated secondary tauopathy [40,41,42]. In the case of both disease hallmarks being present, pioglitazone might have the potential to modulate inflammation caused by amyloidosis, but based on our results it seems to be an ineffective modulator of tau-induced inflammation. This hypothesis is further supported by our immunohistochemistry analysis of AT8-positive cells and Iba1 and CD68 expression, which did not show a difference between long-term treated and untreated P301S mice. We note that future studies could use multiplex panels or single-cell RNA of isolated microglia for a comparison of long-term pioglitazone treatment between Aβ and tau mouse models to elucidate the underlying mechanisms of ineffective treatment in P301S mice.

As a limitation, we did not evaluate wild-type samples in the immunohistochemical analysis, and thus our data do not allow the judgement of quantitative Iba1 and CD68 expression in P301S mice compared to wild-type mice. However, the comparison of P301S and wild-type mice was addressed by TSPO-PET data, where previous studies proved a strong congruency between TSPO-PET and immunohistochemistry [24,25,26]. Therefore, this study was focused on a sensitive comparison of mice with tau pathology instead of performing multiple comparisons with the inclusion of wild-type mice. Moreover, we did not test a dose-dependent effect, while some studies suggest a dose-dependent effect of pioglitazone [21,37].

In conclusion, we found that long-term pioglitazone had no significant effect on microglial activation in P301S mice as measured with TSPO-PET and immunohistochemistry. Our results led to the hypothesis that pioglitazone may have no beneficial impact on tau-mediated neuroinflammation, possibly explaining the failure of this treatment strategy in clinical trials of Alzheimer’s disease. Further studies regarding the mechanistic differences between PPARγ stimulation of Aβ- and tau-related microglial activation will provide novel insights into the pathomechanisms of neurodegenerative diseases and possible new treatment strategies.

## 4. Materials and Methods

### 4.1. Animals and Study Design

Animals were housed in a temperature- and humidity-controlled environment with a 12 h light–dark cycle, with free access to food and water.

TSPO-PET-scans in P301S and wild-type mice were performed at four different time-points, as indicated in Table 1. In the P301S mouse line, which was generated on a C57BL/6 background, the thy1 promoter controlled the expression of mutated human tau. Tau deposits are exclusively found in neurons [27]. These mice showed the first pathological signs of disease as early as three months of age, followed by the formation of neurofibrillary tangles and gliosis of astrocytes and microglia [28,29]. C57BL/6 mice served as wild-type controls. All investigated mice were female.

Cage randomization concerning treatment (pioglitazone) and control (placebo) chow was initiated after the baseline PET scans, and treatment was continued until perfusion of the animals for a total time of 5.5 months. Food pellets of treatment chow contained pioglitazone at a dose of 350 mg per kg of chow [24]. Assuming 5 g food intake per mouse per day, a mouse with a body weight (BW) of 25 g received a dose of ~70 mg/kg-BW. For transcardial perfusion with PBS, mice were deeply anesthetized. Harvested brains were fixed in 4% paraformaldehyde (12 h) and stored in PBS for immunohistochemical analyses.

### 4.2. PET Imaging

Radiochemistry, TSPO-PET image acquisition, and image pre-processing were performed as described previously [43,44]. In brief, mice anesthetized with isoflurane were injected with an average dose of 13.6 ± 2.0 MBq of [^18^F]GE-180. Then, 60 min post-injection, TSPO-PET recordings were performed for 30 min, leading to an emission window of 60–90 min. P301S and wild-type mice were examined simultaneously in a four-mouse chamber imaging setting irrespective of their genotype and treatment in a randomized way. This procedure ensured an equal level of isoflurane anesthesia throughout the whole imaging procedure.

### 4.3. PET Image Analysis

All image analysis were performed using PMOD (version 3.5, PMOD Technologies, Zurich, Switzerland), as described earlier [44].

Different methods of intensity normalization were used to compare TSPO-PET findings with all commonly applied approaches. Therefore, we assessed the cerebral TSPO-expression after SUV-normalization, myocardial correction [45], intracerebral reference-based SUV ratios (SUVR), and injected dose-adjustment (%ID). For reference region normalization, we used the previously validated nucleus accumbens scaling for the generation of standardized uptake value ratios (SUVR) [34]. Furthermore, myocardium-adjusted standardized uptake values (SUV), SUV, and %injected dose (%ID) were used to account for radiotracer dosing, body weight, and individual physiological differences between mice. Brainstem, cerebellum, frontal cortex, and hippocampus served as target regions that had previously been shown to be particularly relevant in this mouse model [26].

### 4.4. Immunohistochemistry

Immunohistochemistry was performed to assess the number of tau-positive cells in the brains of P301S mice. To this end, paraformaldehyde-fixed 50 µm thick sagittal brain sections were incubated for 48 h in PBS with 1% BSA, 5% normal goat serum, and 0.3% Triton X-100 containing mouse monoclonal phosphor-tau primary antibody (Ser202, Thr205 (AT8), 1:1000, Thermo Fisher Scientific Inc., Waltham, MA, USA, MN1020). Afterwards, slices were incubated for 4 h at room temperature with a suitable secondary antibody. Imaging was performed on a confocal microscope (LSM 780 Axio invers, Carl Zeiss AG, Jena, Germany) with a ×20 objective in three sagittal sections. Target areas were selected based on the PET results and consisted of the cortex and the brainstem. Images were processed with the ZEN 3.1 software and image analysis was performed with FiJi/ImageJ [46] by counting the number of tau-positive neurons in the target areas of each section.

To assess the degree of activation of microglia, paraformaldehyde-fixed 50 µm thick sagittal brain sections were incubated overnight at 4 °C in PBS with 5% normal goat serum and 0.5% Triton X-100 containing guinea pig monoclonal anti-Iba1 primary antibody (1:500, Synaptic Systems GmbH, Göttingen, Germany, 234308) and rat monoclonal anti-CD68 primary antibody (1:500, FA-11, Bio-Rad Laboratories Inc., Hercules, CA, USA, MCA1957). Afterwards, slices were washed three times with PBS supplemented with 0.5% Triton X-100, and subsequently slices were incubated for 2 h at room temperature with a suitable secondary antibody. Imaging was performed on a wide-field microscope (Zeiss Axio Vert A1 with ApoTome, Carl Zeiss AG, Jena, Germany) with a ×20 objective in three sagittal sections. Target areas were selected based on the PET results and consisted of the cortex and the brainstem. Images were processed with the ZEN 3.1 software and image analysis was performed with FiJi/ImageJ [46] by quantifying the area with a signal over a certain threshold for Iba1 and CD68.

### 4.5. Statistics

Relevant group differences (i.e., between genotype or treatment) in longitudinal TSPO-PET data were identified with a mixed-effects model and Tukey’s multiple comparisons test using GraphPad Prism statistical software (version 9.4.1 for Windows, GraphPad Software, San Diego, CA, USA). With the same software, an unpaired *t*-test was used to assess statistically significant differences in the immunohistological data. A threshold of *p* < 0.05 was considered significant to reject the null hypothesis.

## Figures and Tables

**Figure 1 ijms-24-10106-f001:**
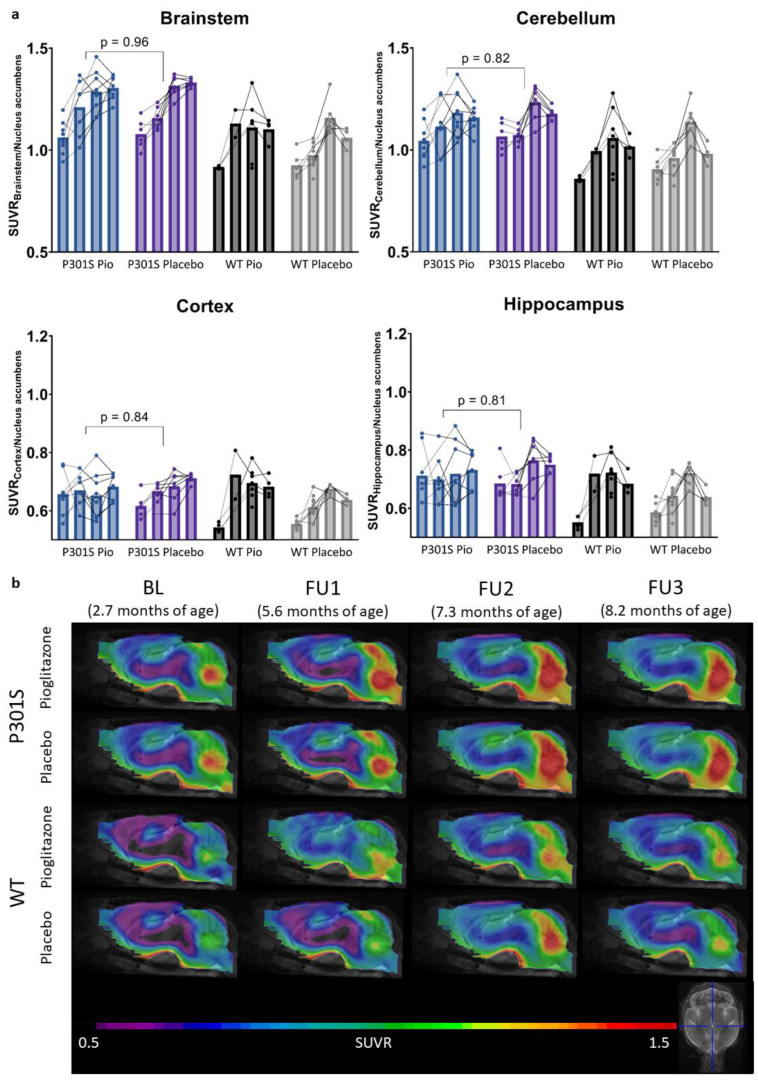
TSPO-PET signal of P301S mice treated with pioglitazone (P301S Pio) or placebo chow (P301S Placebo) and the respective wild-type (WT) control groups over time (BL = baseline, FU = follow-up). (**a**) Individual time courses of TSPO-PET signals in brainstem, cerebellum, cortex, and hippocampus. *p*-Values derive from a *t*-test comparing P301S mice with pioglitazone and placebo treatment independently of the time point. (**b**) Group level TSPO-PET images of pioglitazone or placebo-treated P301S and wild-type mice are shown as sagittal slices upon an MRI template. Data were normalized by average value in nucleus accumbens (SUVR). Extracerebral regions and the pituitary gland were masked.

**Figure 2 ijms-24-10106-f002:**
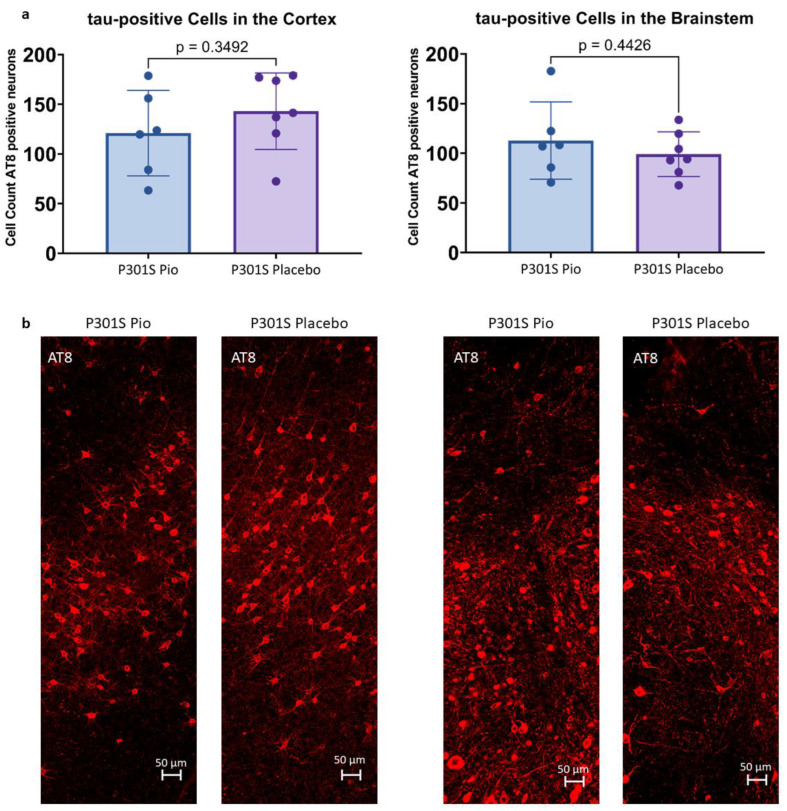
(**a**) Abundance of tau-positive cells in 0.014 mm^3^ of the cortex and 0.011 mm^3^ of the brainstem of P301S mice treated with pioglitazone (P301S Pio) or placebo (P301S Placebo) as counted from triplicates of immunohistological stainings with AT8. (**b**) Representative orthogonal projections of AT8 immunohistochemistry for the cortex (left panel) and the brainstem (right panel) of two P301S mice treated with pioglitazone and placebo, respectively.

**Figure 3 ijms-24-10106-f003:**
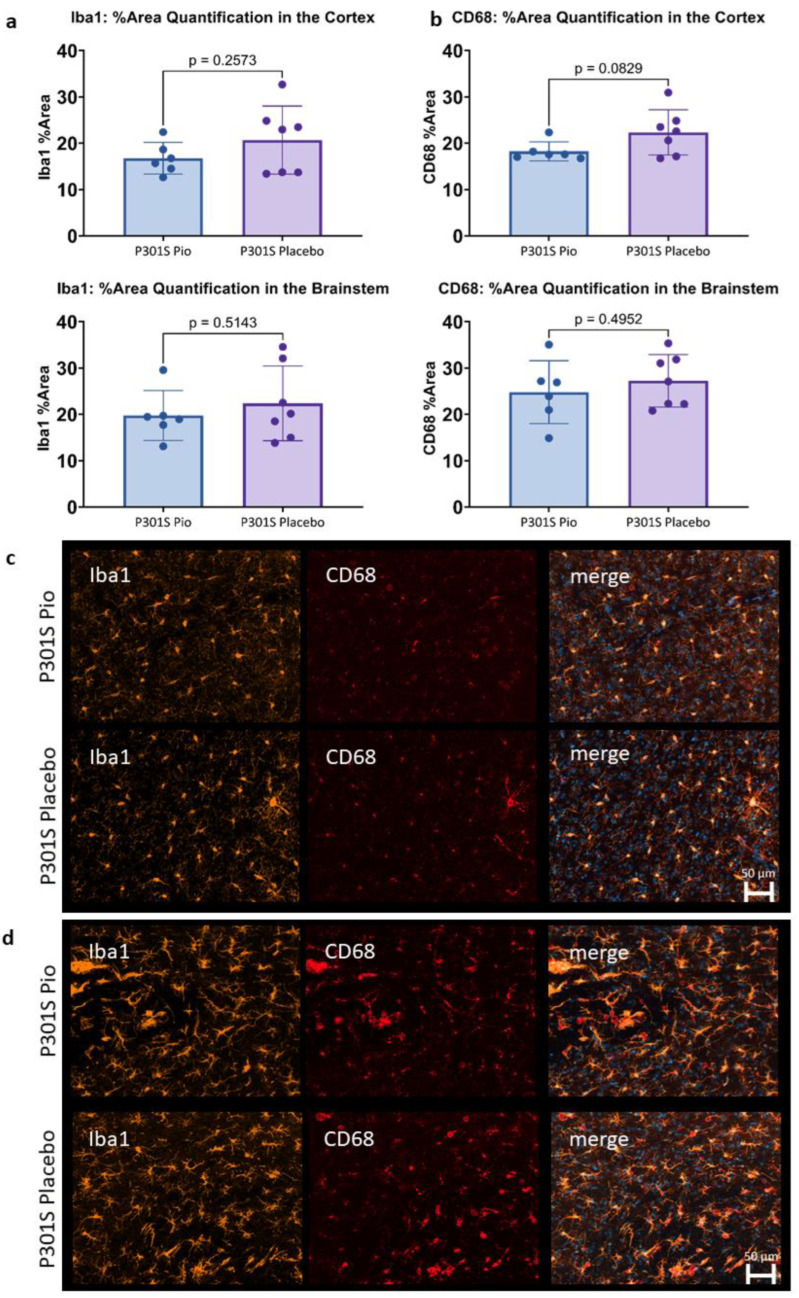
(**a**,**b**) Microglial activation (area%) in the cortex and the brainstem of P301S mice at 8.2 months of age treated with pioglitazone (P301S Pio) or placebo (P301S Placebo) as quantified by analyzing triplicates of immunohistochemical stainings with Iba1 (**a**) and CD68 (**b**). (**c**,**d**) Representative orthogonal projections of immunohistochemical stainings of P301S mice treated with pioglitazone (P301S Pio) or placebo (P301S Placebo) in the cortex (**c**) and the brainstem (**d**).

**Table 1 ijms-24-10106-t001:** Age in months (Age) and number (N) of the mice with successful TSPO-PET for each group (P301S mice with pioglitazone treatment (P301S + Pio), P301S mice with placebo treatment (P301S + Placebo), wild-type mice with pioglitazone treatment (WT + Pio), and wild-type mice with placebo treatment (WT + Placebo), as well as number of mice available for immunohistochemistry (IHC)). f = female.

	BL Age		FU1 Age		FU2 Age		FU3 Age		IHC N
		**N**		**N**		**N**		**N**	
P301S + Pio	2.6	8 (f)	5.9	7 (f)	7.3	8 (f)	8.2	8 (f)	6 (f)
P301S + Placebo	2.6	6 (f)	6.0	6 (f)	7.3	7 (f)	8.2	7 (f)	7 (f)
WT + Pio	2.8	4 (f)	5.9	2 (f)	7.7	6 (f)	8.2	4 (f)	0
WT + Placebo	2.9	6 (f)	5.0	10 (f)	7.4	6 (f)	8.4	4 (f)	0

## Data Availability

All raw data can be obtained from the corresponding author upon reasonable request.

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
