# Peer review of "Long-Term Pioglitazone Treatment Has No Significant Impact on Microglial Activation and Tau Pathology in P301S Mice"

_ijms, 2023, doi:10.3390/ijms241210106_

Round 1

Reviewer 1 Report

The authors investigate the ability of pioglitazone to ameliorate the levels of inflammation associated with the progression of neurodegeneration over time. However, there are some significant issues which need to be addressed for proper evaluation of the current findings. 

1. Please dictate the dose of pioglitazone, how it was delivered, how many days all pertinent information including what was it reconstituted in. 

2. There is wild type data with and without drug in some experiments and missing in others. Why is this? it is difficult to interpret the analysis in figs. 2 and 3 because there is no wild type to compare what is baseline. 

3. The aspect of changes in Tau levels. Is this phosphorylated Tau or total Tau? If it is changes in total Tau what does this mean for the microtubule stability or architecture? If Tau is reduced does this mean an increase in NFTs or reduced axonal integrity? 

4. Is there any secondary or alternative forms of measurement of TSPO, Tau, CD68 or IBA1levels in the different regions of the brains such as western analysis or qPCR? 

5. Please add why is investigating the brain stem and other areas of the brain significant for Alzheimer's disease, as your animal model is a model of AD. 

Quality of English including syntax is fine. it is well written. 

Reviewer 2 Report

In the MS written by Kunze et al. authors investigated the effect of pioglitazone (a PPARg ligand) on a neurodegenerative disease (taupathy) model on P301S transgenic and on control animals. The study is based upon the  translocator protein positron-emission-tomography (TSPO-PET) imaging and immunohistochemistry technigues to assess microglial activation  together with terminal AT8 immunohistochemistry for Tau protein detection. All of the tests were negative, the pioglitazone treatment did not influence the pathological Tau accumulation in P301S transgenic animals.

General comments

The results are adequate, the interpretation is based upon the experimental data.

However some parts of the MS need further attention.

 Major points

1) The introduction is too short and does not describe basic background information.

55-57 „In earlier studies, pioglitazone has been shown to be a promising treatment for Alzheimer’s disease, ameliorating both the pathology as well as the cognition in animal models.” – references are required.

59-61 „Based on previous findings in Aβ mouse models, we tested the hypothesis that decreased microglial activation is detectable by serial 18 kDa translocator protein positron-emission-tomography (TSPO-PET) in pioglitazone-treated P301S mice.” – authors ought to explain the connection between the TSPO-PET and the microglial activation.

62 “P301S mice” – authors should explain what is and why is P301S mouse strain a good choice to study in the current project.

2) Results

The description of the results is not always clear.

75-76 „Similar results were found for SUV normalized data, myocardium adjusted SUV or %ID  (Figure A1-A3). „ What was the purpose of the adjustment for myocardium? Why for myocardium? When was the Pio treatment started?

Fig.2. How were the microscopic results evaluated? It would be nice to see how many percentage of the cells was AT8 positive.

194thy1 promotor” possible typo?

118-123 Figure 3. At what age were immunohistochemical stainings done? T

Figure legend A1 and A2 FU and BL baseline and follow up –however on the graph neither the FU nor the BL was found. It seems to be unnecessary to use these abbreviations in the Figure legend.

Discussion is adequate and well-written.
